# Changes in Soil C:N:P Stoichiometry and Microbial Structure along Soil Depth in Two Forest Soils

**Lei Hu [1,2], Luji Ade [2], Xinwei Wu [1], Hongbiao Zi [2], Xueping Luo [2] and Changting Wang [2,\*]**

[1]    School of Life Sciences, Nanjing University, Nanjing 210046, China; hl007873@163.com (L.H.); xinwei_8008@126.com (X.W.)

[2]    Institute of Qinghai-Tibetan Plateau, Southwest Minzu University, Chengdu 610041, China; luji1991@163.com (L.A.); zhbyn2010@163.com (H.Z.); luoxpchongqing@163.com (X.L.)

\*    Correspondence: wangct@swun.edu.cn; Tel.: +86-136-8803-6318

**Abstract:** The effects of interactions of soil type and soil depth on soil C:N:P stoichiometry and microorganisms are poorly understood. In this study, soil samples (0–10, 10–20, 20–30, 30–50, 50–100 cm) were collected from two soil types (Haplic luvisols and Eutric cambisols) in *Sabina przewalskii* Kom. forest of Qinghai-Tibet Plateau. The soil C:N ratio and soil microbial biomass (SMB) measured using phospholipid fatty acid in Eutric cambisols were significantly higher than in Haplic luvisols, while soil C:P and N:P ratios were the opposite. In the two soil types, the soil C:N ratio significantly increased with soil depth, and the soil C:P and N:P ratios declined. Structural equation modelling (SEM) indicated that soil depth directly affected soil C, N and P contents. Soil type and soil depth could directly affect soil fungal and bacterial biomass, and indirectly affect both of them through soil bulk density. Meanwhile soil fungal biomass was influenced by soil depth through Total C. These results highlighted that the vertical distribution of soil bacteria could largely be attributed to changes of soil fungi depending on soil carbon resources.

**Keywords:** soil type; soil profile; ecological stoichiometry; phosphor lipid fatty acid; structural equation model

## 1. Introduction

Soil carbon (C), soil nitrogen (N), and soil phosphorus (P) contents and their stoichiometry are important indicators of soil organic composition and soil quality, contributing to understand soil C storage, C sequestration, biogeochemical cycles, ecosystem succession and degradation, and the response to global climate change [1]. In general, the C:N ratio is negatively correlated with its decomposition rate, and a lower C:N ratio demonstrates a quick demineralization process. Soil C:N ratios are often considered a constant (10:1) in estimating global soil carbon storage [2]. Soil C:N ratio may explain 99.2% of variation in soil dissolved organic carbon and may therefore be used to explore organic matter decomposition in riverine soils [3]. Soil N:P ratios are used to indicate N, and P saturation and quantify the threshold of nutrient limitation [4]. Limited N:P ratios would result in elevating mineralization enzyme activity and affect decomposition rate of soil organic C and the quality and quantity of litter [5]. Soil microbial biomass C:P ratio is predicted to balance P release from soils and P intake from environments [6].

There are many factors affecting the contents of soil C, N, P, and their stoichiometry, such as geographic position, soil type, vegetation types, soil fauna, and land use. There were a wide range of C:N:P ratios when comparing data from studies in different counties [2]. Previous studies have also demonstrated the significant difference of the soil C:N:P stoichiometry among different soil groups or ecosystem types [7,8]. Moreover other studies explored the effects of land use types and soil depth on

particle size and soil organic C density in subtropical China [9], and confirmed that soil C-N coupling was irrespective of ecosystem type and soil depth [10].

Soil quality does not depend only on the physicochemical properties of soil, but is also very closely linked with the biological properties of soil mostly in microbiological processes [11]. Among the indigenous soil components, the soil microbial community plays a key role in different ecologically important functions in the soil. It is involved in litter decomposition, the cycling of nutrients, the formation of stable microaggregates, and structural development. Many studies have shown that the soil microbial community structure and biomass varied along environmental gradients because of plant traits or soil abiotic factors [12,13]. Bacteria and fungi are the main soil microbial groups, and both of them functioned in soils to decompose organic material and affected soil nutrients. Additionally, soil microbial biomass was assessed and determined to be the most important soil ecological characteristic [14]. Changes in the soil microbial biomass C:N ratio resulting from the bacteria-fungi ratio are thought to be important in explaining different N mineralization patterns [15]. The soil organic C and total N were consistent with the spatial patterns of soil microbial biomass C and N [16].

Overall the soil C:N:P stoichiometry and soil microbial characteristics were significantly correlated with the abiotic environments and anthropogenic activities. Even in the same ecosystem type, the soil stoichiometry characteristics might also be influenced by different soil type [17]. Along the Qilian Mountains, in the northeast of the Qinghai-Tibet Plateau, there existed two forest soil types in the same forest type dominated by Qilian juniper (*Sabina przewalskii* Kom.), which mainly resulted from different mountain climatic conditions. In the Qilian juniper forests, Haplic luvisols in the middle of the Qilian Mountains was influenced by the arid climate and weak clayification, while in the east of the Qilian Mountains the warm humid climate resulted in Eutric cambisols [18]. In order to adequately explore whether changes in soil C:N:P stoichiometry and soil microorganisms of two soil types in the same Sabina przewalskii forest were also similar, we (1) compared the differences in the soil C:N:P stoichiometry with soil depth in two forest soils; (2) studied the changes in soil fungal and bacterial biomass; (3) demonstrated the relationships between soil C, N, and P contents and soil microbial biomass.

## 2. Material and Methods

### 2.1. Study Site

Our study area included a *Sabina przewalskii* forest in the Qinghai-Tibet Plateau, which accounts for approximately one-quarter of China's total land. The Qinghai-Tibet Plateau is the highest, largest and youngest in the world, and it experiences a Plateau dry continental climate.

We conducted this study in the Dulan County (36°18′ N, 98°9′ E) in Qinghai Province, China. The average annual temperature is approximately 3 °C, with maximum and minimum temperatures of 14.9 °C in July and −10.2 °C in January, respectively. The annual precipitation is about 180–200 mm.

The experimental site represents the middle stage of the natural coniferous forest. The dominant tree species is Qilian juniper (*Sabina przewalskii*), which is a unique species endemic to China. It lives in very cold, soil poor and arid regions. According to the field study, the canopy density is about 45%–65%, and the community height is about 7.5–9.5 m. The dominant genus in the shrub and herb layer is *Polygonum* and *Elymus*, respectively. The biomass of litter is about 200 g·m$^{-2}$, and the biomass of fine roots at 0–40 cm is about 3.8 g·cm$^{-2}$.

### 2.2. Experimental Design and Field Sampling

In July 2013, three forest plots in the yellow-brown earths (Haplic luvisols, HaL#1, #2, #3) and cinnamon earths (Eutric cambisols, EuC#1, #2, #3) were chosen in Dulan County, and the experimental sites were in Table 1. In order to differentiate soil yellow-brown earths and brown earths, soil profile (0–100 cm) and soil genetic layer were explored in the 6 sites, respectively. According to the following

criterion, we divided soils at 0–100 cm into the O layer, A layer and B layer. Soils of the O layer were mainly filled with decomposed litters. Soils of A layer were mineral soils under O layer. It was often accumulated with a humification organic layer intensively mixed with mineral substances. The soils at the B layer almost lost the rocky structure, and (1) accumulated with silicate clay particles, humus, carbonate, or $SiO_2$; (2) dissolution of carbonates; (3) were beneficiated with trioxide or dioxide; (4) lower brightness, higher color, and reddish hue resulting from trioxide or dioxide cutans; (5) granular, blocky or prismatic structure. The O, A, and B layers in Haplic luvisols were at 0–3 cm, 3–16 cm, and 16–100 cm depths, respectively. In Eutric cambisols, the O, A, and B layers were 0–2 cm, 2–10 cm and 10–100 cm. The transition layer in Haplic luvisols was horizontal, and reached 2–5 cm, while the transition layer in Eutric cambisols formed a recessed region, and the thickness of transition layer reached over 12 cm. Observed following the Munsell Color Chart, the Haplic Luvisols at A layer was 5YR2/1 and at B layer was 2.5Y5/4. The Eutric cambisols at A layer was 10YR4/5 and at B layer was 7.5YR4/6.

**Table 1.** Geographical and plant community characteristics in *Sabina przewalskii* Kom. forest.

| Sample Plot | Geographical Location | Altitude (m) | Slope (°) | Canopy Density (%) | Mean Tree Height (m) | Mean Forest DBH (cm) | Number of Trees | Forest Age (year) | Plot Size (m²) |
|---|---|---|---|---|---|---|---|---|---|
| HaL#1 | 36°01′48″ N 98°11′38″ E | 3796 | 33 | 50 | 8.07 | 28.2 | 112 | 30 | 600 |
| HaL#2 | 36°04′01″ N 98°13′39″ E | 3798 | 31 | 45 | 6.72 | 21.9 | 111 | 30 | 600 |
| HaL#3 | 36°04′03″ N 98°13′32″ E | 3778 | 31 | 45 | 9.54 | 25.1 | 88 | 30 | 600 |
| EuC#1 | 35°59′49″ N 98°06′10″ E | 3619 | 36 | 65 | 9.98 | 25.5 | 107 | 40–50 | 600 |
| EuC#2 | 35°59′18″ N 98°06′37″ E | 3718 | 36 | 45 | 11.37 | 31.4 | 110 | 40–50 | 600 |
| EuC#3 | 35°59′16″ N 98°06′33″ E | 3726 | 37 | 45 | 10.32 | 28.6 | 92 | 40–50 | 600 |

HaL#1, HaL#2, and HaL#3 means the three forest plots in Haplic luvisols. EuC#1, EuC#2, and EuC#3 means three forest plots in Eutric cambisols. DBH = Diameter at the breast height.

The dimensions of the experimental areas were 20 × 30 m². Firstly, all the Qilian junipers were measured to acquire the tree height and diameter of breast height arborous layer; the coverage, the height, and the biomass of shrub were also measured at the shrub layer; and then the aboveground plant biomass was harvested at the herb layer.

At each plot (30 m × 20 m), three 1 m × 1 m subplots were chosen for aboveground plant community survey. After removal of aboveground herb plants, five soil samples were collected from 0–10, 10–20, 20–30, 30–50, and 50–100 cm along soil profile, respectively. At each soil layer, five soil samples were collected from one subplot in a V shape using a cylindrical core (90 cm in depth and 5 cm in diameter). The soil samples were mixed and then divided into two parts. One part was stored in the freezer at −20 °C for soil microbial composition using phospholipid fatty acid analysis (PLFAs), and the other part was stored at 4 °C for soil nutrient analysis.

*2.3. Soil Physicochemical Properties*

Five cutting rings (inner diameter 5.03 cm, volume 100 cm³) filled with soil along soil profiles were used to weigh the samples, and then were dried at 105 °C to re-weighed to obtain gravimetric soil water content and soil bulk density.

The soil samples stored at 4 °C were air-dried, ground and sieved through a 2-mm mesh to measure their chemical properties. Soil total carbon (TC), nitrogen (TN) and phosphorus (TP) contents were measured by an elemental analyzer (Flash EA 1112 series, CE Instruments, Milan, Italy) [19].

### 2.4. Soil Microbial Biomass

Soil microbial community composition of each soil sample was assessed by measuring the microbial phospholipid fatty acid (PLFA) composition according to Bardgett and Frostegård [20,21]. Fatty acid nomenclature was used as previously described [21]. Based on previous results and using PLFA biomarkers, the present study showed that the appraised carbon chain length of PLFA was C14 to C20 for rich species, including bacteria and fungi, of which there were 20 types in total. Soil bacteria were represented by PLFAs C13:0, C14:0, C15:0, C16:0, C18:0, 16:1ω7c, 16:1ω9c, 18:1ω7c, cy17:0, cy19:0, i15:0, i16:0, i17:0, i18:0, 2Me18:0, and 10:Me18:0. Soil fungi were represented by 18:1ω9c and 18:2ω6,9 [22–25].

### 2.5. Data Analysis

First, the six plots (HaL#1, #2, #3, and EuC#1, #2, #3) and soil samples at different soil depths (0–10, 10–20, 20–30, 30–50, and 50–100 cm) were considered as the random factors, and a linear mixed-effects model was used to test the random effects. Then the fixed effects of soil type and soil depth on soil physical properties, soil C:N:P stoichiometry and soil microbial biomass were analyzed with multivariate analysis of variance (ANOVA) with LSD (Least Significant Difference) tests at the significance levels of $p = 0.05$ and 0.01, which were performed using R v3.5.0. Structural equation modelling (SEM) was used to analyses different hypothetical pathways that might explain the direct effects of soil type and soil depth on soil C, N, and P contents, and soil microorganisms and their indirect effects through soil bulk density, soil water content, which was performed using Amos 23.0 (Amos Development, Spring House, Armonk, PA, USA). In order to avoid the mutual linear problem, the soil C, N and P contents were analyzed in the SEM instead of the soil C:N, N:P and C:P ratios.

## 3. Results

### 3.1. Soil Physical Properties

Soil type, soil depth and their interaction significantly affected the soil bulk density, but only soil type had significant effects on soil water content. Soil bulk density was significantly affected by soil type, soil depth and their interaction, but soil water content was not significantly influenced (Table 2). In the two forest soils, the soil bulk density significantly increased with soil depth, while in the same soil depth the soil bulk density of Haplic Luvisols was always significantly higher than that of Eutric cambisols. Instead, the soil water content did not change significantly along soil depth in the Haplic Luvisols soils, and in the Eutric cambisols soils it significantly increased from 0–30 cm to 30–100 cm. There was no significant difference between the same soil depths of the two forest soils. Overall, the soil bulk density and water content of Eutric cambisols were higher than that of Haplic Luvisols (Figure 1).

**Table 2.** Analysis of variance for soil physicochemical properties in two forest soil types.

| Variable | Soil Type (ST) | | Soil Depth (SD) | | ST × SD | |
|---|---|---|---|---|---|---|
| | *F* | *p* | *F* | *p* | *F* | *p* |
| Soil bulk density (g·cm$^{-3}$) | 120.810 | <0.001 ** | 3.671 | 0.026 * | 8.705 | <0.001 ** |
| Soil water content (%) | 1.898 | 0.240 | 2.516 | 0.083 | 0.549 | 0.702 |
| C:N ratio (C/N) | 19.221 | 0.012 * | 23.414 | 0.001 ** | 2.149 | 0.122 |
| C:P ratio (C/P) | 35.268 | 0.004 ** | 11.222 | <0.001 ** | 7.423 | 0.001 ** |
| N:P ratio (N/P) | 23.721 | 0.008 ** | 22.612 | <0.001 ** | 7.799 | 0.001 * |
| Soil microbial biomass (nmol·g$^{-1}$) | 54.546 | 0.002 ** | 220.926 | <0.001 ** | 24.746 | <0.001 ** |
| Soil bacterial biomass (nmol·g$^{-1}$) | 50.365 | 0.002 ** | 84.992 | <0.001 ** | 9.625 | <0.001 ** |
| Soil fungal biomass (nmol·g$^{-1}$) | 1.798 | 0.251 | 20.853 | <0.001 ** | 5.153 | 0.007 ** |

* and ** indicate significant ($p < 0.05$) and highly significant correlation ($p < 0.01$), respectively.

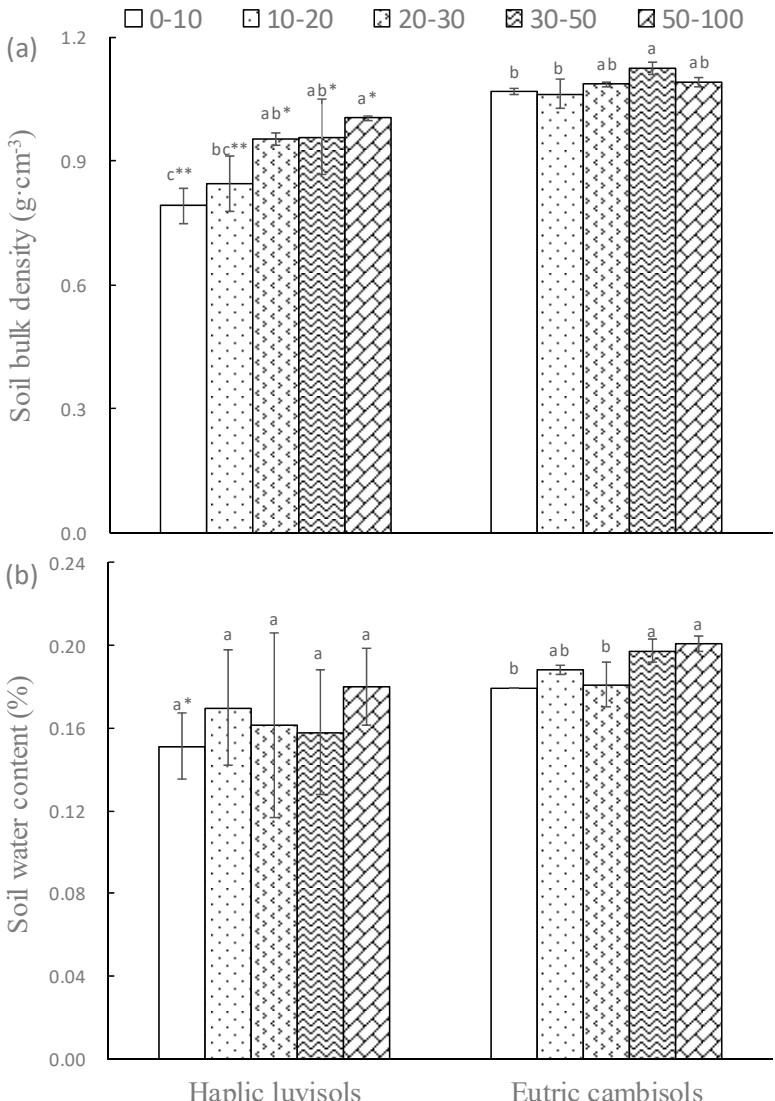

**Figure 1.** The soil bulk density bulk and water content in the Haplic luvisols (**a**) and Eutric cambisols (**b**). Different lowercase letters indicated a significant difference among soil depth in the same forest soil; * and ** indicates significant ($p < 0.05$) and extremely significant ($p < 0.01$) differences between the two forest soils in the same soil depth, respectively. The same below.

### 3.2. Soil C:N:P Stoichiometry

Except for soil C:N ratio, the ANOVA performed over the data set of soil chemical variables all exhibited a highly significant overall interaction between soil type and soil depth (Table 2). Soil C:N ratio increased with soil depth in the two forest soils, and reached the maximum at 50–100 cm. Meanwhile, soil C:N ratio in Eutric cambisols soils were still higher than that of Haplic Luvisols soils, especially at 10–30 cm it reached a significant level. In contrast, the C:P and N:P ratio of Haplic Luvisols soils were higher than that of Eutric cambisols soils, especially at 0–30 cm. Moreover, the C:P and N:P ratio of Haplic Luvisols soils decreased significantly with soil depth, while in the Eutric cambisols soils they did not significantly change with soil depth, especially for the soil C:P ratio (Figure 2).

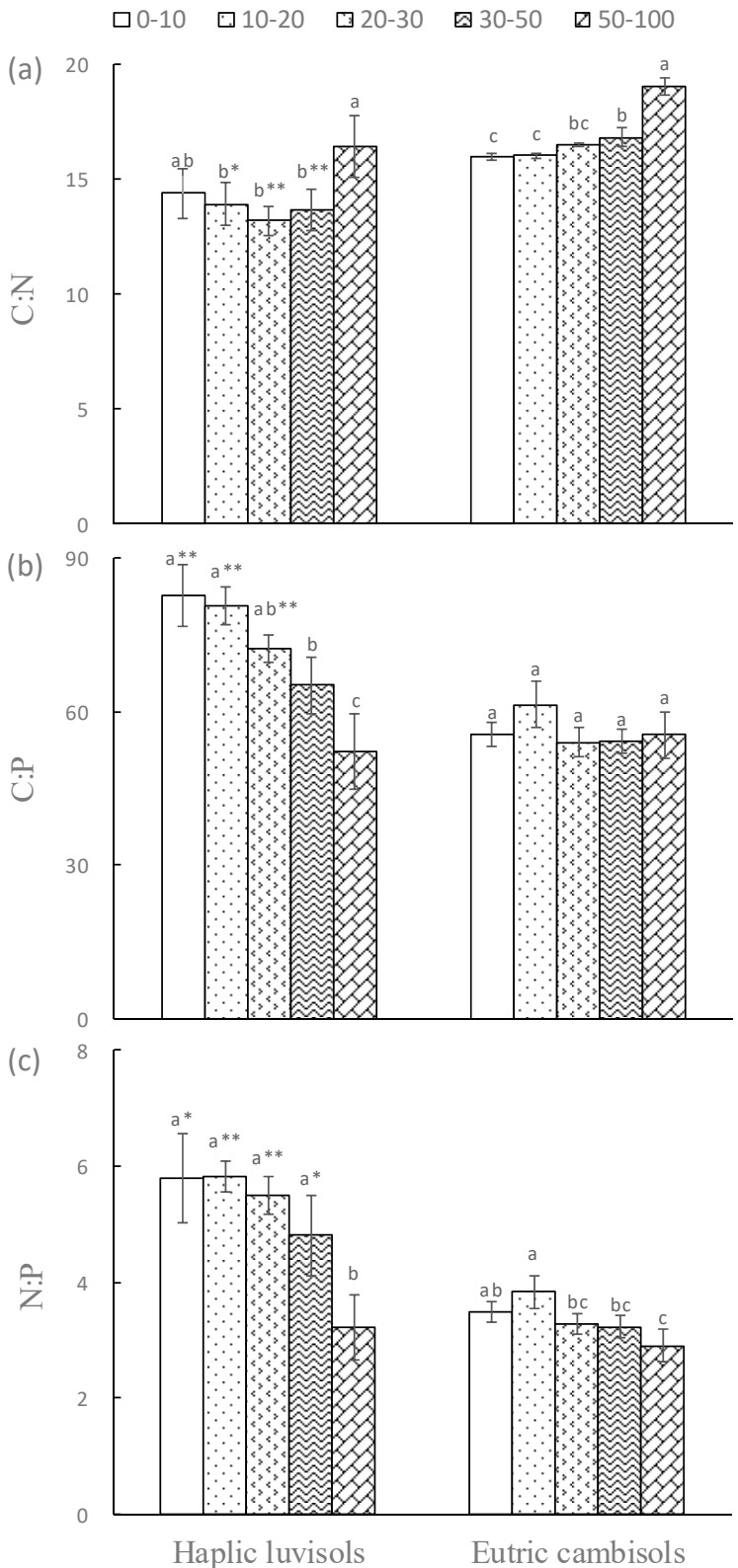

**Figure 2.** Changes in soil C:N:P stoichiometry in Haplic luvisols and Eutric cambisols.(**a**) C:N ratio, (**b**) C:P ratio, (**c**) N:P ratio.

### 3.3. Soil Microbial Properties

Soil type, soil depth and their interaction had significant effects on soil microbial biomass and soil bacterial biomass, but there was no effect of soil type on soil fungal biomass (Table 2). Compared with

the Eutric cambisols soils, the soil microbial biomass, soil bacterial biomass and soil fungal biomass were much lesser, especially significantly lower at 0–10 cm. The similarities between the two forest soils were that soil microbial biomass, bacterial biomass, and fungal biomass significantly declined with soil depth (Figure 3).

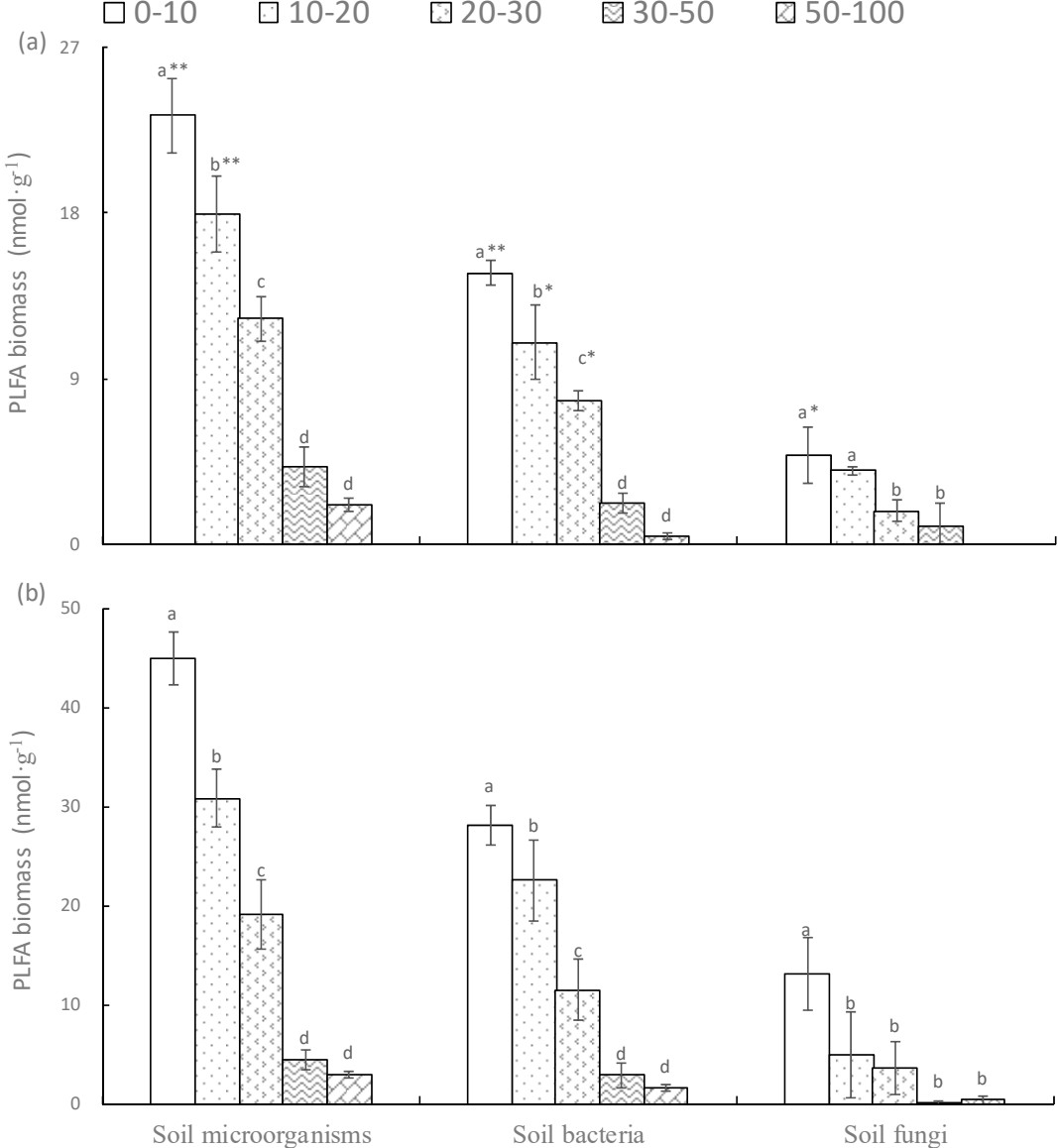

**Figure 3.** The soil microbial biomass in Haplic luvisols (**a**) and Eutric cambisols (**b**). PLFA means phosphor lipid fatty acid.

### 3.4. The Effects of Soil Type and Soil Depth on Soil Physiochemical Properties and Soil Microbial Biomass

As shown in the SEM model, the direct path from soil type to soil bulk density (SBD) or soil water content (SWC) was stronger than that from soil depth, and the soil type and soil depth explained 68% and 31% of the variation in the SBD and SWC, respectively (Figure 4 and Tables 3 and 4).

Soil type had no significant direct effects on TC, TN and TP, while soil depth had about equally significant direct paths to TC (−0.461) and TP (−0.472), which were less than TN (−0.572). Both soil type and soil depth could indirectly affect TC and TN through SBD. The total (direct and indirect) effects of soil type and soil depth explained 40%, 55%, and 43% of the variation in TC, TN and TP, respectively (Figure 4 and Tables 3 and 4).

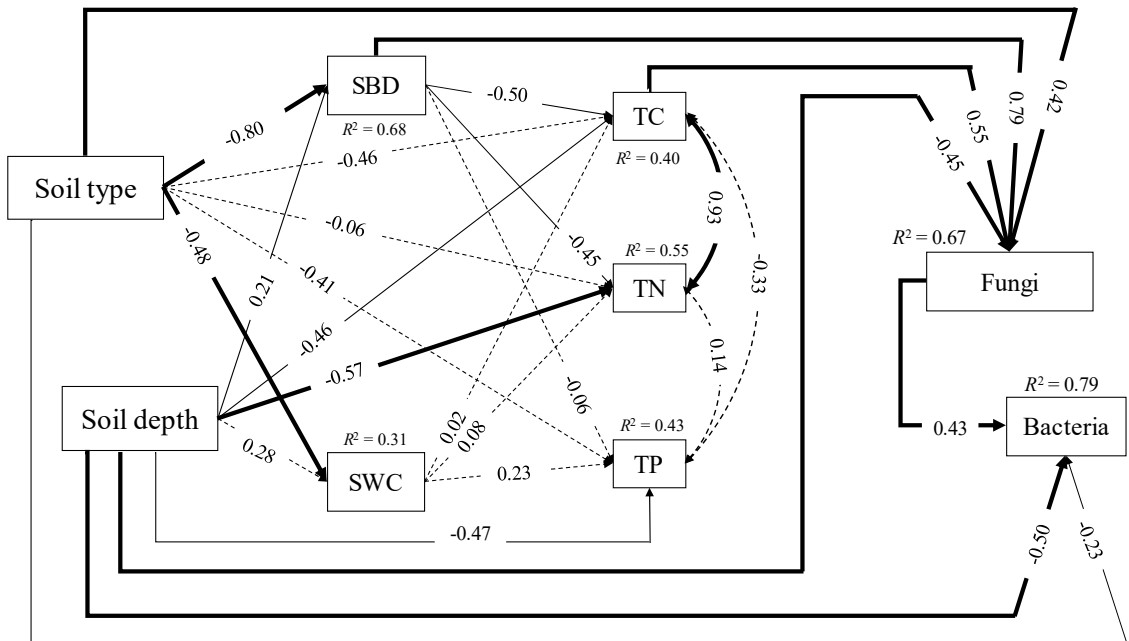

**Figure 4.** Structural equation modeling on the direct and indirect effects of soil type and soil depth on soil bulk density (SBD) and soil water content (SWC) on the contents of soil total carbon (TC), total nitrogen (TN), and total phosphorus (TP), and soil microbial biomass (soil bacterial biomass and fungal biomass). The final model fit the data well: $\chi^2 = 8.193$, df = 10, GFI = 0.946, RMSEA < 0.001, $p = 0.610$. Numbers at solid arrows ($p < 0.05$) were standardized path coefficients, and width of the arrows indicated the strength of the relationships. The dashed arrows indicated nonsignificant relationships ($p > 0.05$). Numbers close to variables ($R^2$) indicated the variance explained by the model. Here df, GFI and RMSEA mean degree of freedom, goodness of fit index, root-mean square error approximation, respectively.

**Table 3.** Regression weights and covariance in the structural equation model.

| Response Variables | | | Estimate | Standard Error | $p$ |
|---|---|---|---|---|---|
| SBD | <— | Soil type | −0.199 | 0.026 | <0.001 ** |
| SBD | <— | Soil depth | 0.001 | 0.000 | 0.046 * |
| SWC | <— | Soil type | −0.025 | 0.008 | 0.002 * |
| SWC | <— | Soil depth | 0.000 | 0.000 | 0.070 |
| TC | <— | Soil type | −12.532 | 7.091 | 0.077 |
| TC | <— | Soil depth | −0.196 | 0.068 | 0.004 * |
| TC | <— | SBD | −54.070 | 27.502 | 0.049 * |
| TC | <— | SWC | 7.985 | 89.177 | 0.929 |
| TN | <— | Soil type | −0.105 | 0.432 | 0.808 |
| TN | <— | Soil depth | −0.017 | 0.004 | <0.001 ** |
| TN | <— | SBD | −3.421 | 1.675 | 0.041 * |
| TN | <— | SWC | 2.972 | 5.430 | 0.584 |
| TP | <— | Soil type | −0.062 | 0.038 | 0.107 |
| TP | <— | Soil depth | −0.001 | 0.000 | 0.002 * |
| TP | <— | SBD | −0.039 | 0.149 | 0.793 |
| TP | <— | SWC | 0.657 | 0.483 | 0.174 |
| Bacteria | <— | Soil type | −4.166 | 1.631 | 0.011 * |
| Bacteria | <— | Soil depth | −0.143 | 0.031 | <0.001 ** |
| Bacteria | <— | Fungi | 0.906 | 0.236 | <0.001 ** |
| Fungi | <— | Soil type | 3.584 | 1.659 | 0.031 * |
| Fungi | <— | Soil depth | −0.060 | 0.017 | <0.001 ** |
| Fungi | <— | TC | 0.172 | 0.043 | <0.001 ** |
| Fungi | <— | SBD | 26.899 | 6.814 | <0.001 ** |

SBD: Soil bulk density; SWC: Soil water content; TC: Total soil carbon contents; TN: Total soil nitrogen contents; TP: Total soil phosphorus content. * and ** indicates significant ($p < 0.05$) and extremely significant ($p < 0.01$).

**Table 4.** Standardized direct, indirect and total effects on total soil C, N, P contents and soil microbial biomass.

| Factors | Effects | SWC | SBD | TC | TN | TP | Bacteria | Fungi |
|---|---|---|---|---|---|---|---|---|
| | Direct | −0.480 | −0.797 | −0.462 | −0.055 | −0.409 | −0.229 | 0.419 |
| Soil type | Indirect | | | 0.390 | 0.318 | −0.058 | −0.105 | −0.667 |
| | Total | −0.480 | −0.797 | −0.072 | 0.263 | −0.468 | −0.335 | −0.247 |
| | Direct | 0.280 | 0.210 | −0.461 | −0.572 | −0.472 | 0.502 | −0.448 |
| Soil depth | Indirect | | | −0.100 | −0.071 | 0.050 | −0.251 | −0.141 |
| | Total | 0.280 | 0.210 | −0.561 | −0.643 | −0.422 | −0.754 | −0.590 |
| | Direct | | | 0.015 | 0.082 | 0.229 | | |
| SWC | Indirect | | | | | | 0.004 | 0.008 |
| | Total | | | 0.015 | 0.082 | 0.229 | 0.004 | 0.008 |
| | Direct | | | −0.499 | −0.448 | −0.065 | | 0.787 |
| SBD | Indirect | | | | | | 0.220 | −0.272 |
| | Total | | | −0.499 | −0.448 | −0.065 | 0.220 | 0.515 |
| | Direct | | | | | | | 0.546 |
| TC | Indirect | | | | | | 0.233 | |
| | Total | | | | | | 0.233 | 0.546 |
| Fungi | Direct | | | | | | 0.426 | |
| | Total | | | | | | 0.426 | |

The factors affecting soil fungal and bacterial biomasses were different. Soil type, soil depth, SBD and TC were directly influencing soil fungal biomass, and they can explain 67% of the variation in total. The total effects of SBD and TC on soil fungal biomass were positive and about equal, with scores of 0.515 and 0.546, respectively, while the soil type and soil depth were negative with −0.247 and −0.590 (Table 4). The two paths, from soil depth or soil type to soil bacterial biomass, were direct and significant, with SBD scores of −0.754 and −0.335, respectively, adding the effects of fungi with 0.426, could explain 79% of the variation (Figure 4 and Table 4).

## 4. Discussion

The yellow-brown earths (Haplic Luvisols) were a forest steppe consisting of Qianlian juniper, grass and sedge groups. Under an arid and semiarid climate, Haplic Luvisols was generated through organic matter accumulation and $CaCO_3$ leaching and deposition. The cinnamon earths (Eutric Cambisols) were developed and formed under a warm humid climate and the main pedogenic processes of the cinnamon earths were humus accumulation and weak acidic leaching [18]. This might explain why the soil water content in Eutric Cambisols was higher than that in Haplic Luvisols. Depending on the results of the soil genetic layer, the O and A layer in Haplic Luvisols was deeper than that in Eutric Cambisols, which resulted in a lower SBD in Haplic Luvisols.

The pedogenic processes and climate conditions of Eutric Cambisols supported a higher level of soil organic carbon (SOC), soil microbial biomass, respiration and potential nitrification than those in Haplic Luvisols [11]. Although a higher level of soil C content was found in Eutric Cambisols, more soil microbial biomass could speed up soil mineralization and release more soil N [1]. There was a negative correlation between SBD and the soil C:N ratio [26], which explained that a quick decrease of the soil C:N ratio in Haplic Luvisols resulted from the SBD of Haplic Luvisols soils, which increased more quickly from surface soils to deeper soils than that of Eutric Cambisols. Previous studies showed that the soil C:P ratio was positively correlated with SBD, SWC, and $HCO_3^-$ contents, while it was negatively correlated with >10 °C accumulated temperature, the average annual relative humidity, and the underground biomass [27]. Soil P content depended on the decomposition rate of soil organic matter [1]. Compared with Haplic Luvisols, double soil microbial biomass in Eutric Cambisols resulted in more rapid decomposition rate of soil organic matter and more P contents were released into soils, which resulted in a significantly higher soil C:P ratio in Haplic Luvisols than that in Eutric Cambisols. Additionally, the decreasing soil C:P ratio with soil depth was consistent with soil microbial biomass in Haplic Luvisols, while in Eutric Cambisols the C:P ratio did not change significantly with soil

depth. This might be explained by the pedogenic processes. The various difference between the two forest soils were the leaching and deposition of calcium carbonate resulting in a recessed transition layer in Eutric Cambisols. The recessed transition layer might be the major reason why there were no significant differences in soil depth in Eutric Cambisols. Similar to the soil C:P ratio, the soil N:P ratio at 0–30 cm in Haplic Luvisols was significantly higher than that of Eutric Cambisols which showed the combined effects of soil microbial biomass and the pedogenic processes. In Eutric Cambisols, more soil microbial biomass meant that they would take in more soil N into their bodies, while more soil microorganisms could also release more soil P because of a quicker soil mineralization [1]. These changes in soil C:N:P stoichiometry with soil depth in the two forest soils were also demonstrated in the SEM analysis, which showed that soil depth had direct effects on soil C, N and P contents, while the direct effects of soil type were not significant.

Soil microbial biomass was regulated by both abiotic factors, including temperature, precipitation, soil nutrients and biotic factors, such as plant diversity and biomass [23]. Meanwhile, the successional stage, and land use history could have a significant effect on soil microbial biomass [28]. Haplic Luvisols was generated under an arid and semiarid climate, while Eutric Cambisols was located in a warm humid climate [18]. The warmer and more humid climate in Eutric Cambisols provided a comfortable environment for soil microorganisms, which explained a significantly higher soil microbial biomass in Eutric Cambisols than in Haplic Luvisols [11]. In forest soils, the percentage of SOC at 0–20 cm averaged 50% [29], and fewer available resources for soil microorganisms with soil depth led to less soil microbial biomass, thereby supporting the decrease in the soil microbial, bacterial and fungal biomasses with soil depth in the two forest soils, which were demonstrated in the SEM analysis indicating that the direct effects of soil depth and soil type on soil bacterial and fungal biomass were significant. Soil bacteria and fungi, the two dominant groups of soil microorganism, need soil C as energy resources. Soil fungi, however, were the main decomposer and tend to be more successful in coniferous forest than soil bacteria [30]. Therefore, in the *Sabina przewalskii* forest soils, soil fungi would be more affected by TC. The carbon resources of soil bacteria were largely attributed to the decomposition of soil fungi, especially in deep soils; both soil fungi and bacteria would be limited to decreasing C resources, while soil fungi preferred relatively infertile soils [31], which resulted in soil bacteria being more affected by soil fungi.

## 5. Conclusions

This study indicated that soil C:N:P stoichiometry was different in the two forest soils, and would change with the soil profile. The difference in pedogenic processes and soil depth largely determined soil C:N:P stoichiometry. Having stronger resource-decomposition in coniferous forests, thus the soil fungi were more affected by soil type while soil bacteria were significantly affected by soil depth. Our results highlighted that the drivers of soil bacteria included not only soil depth but also soil fungi in the coniferous forest.

**Author Contributions:** L.H. and L.A. designed the experiments; X.L. and H.Z. mainly analyzed the data; L.H. and L.A. wrote the original draft; L.H., X.W. and C.W. reviewed and revised the manuscript.

**Funding:** This work was supported by the National Natural Science Foundation of China (31870407, 31370542), the Key Research and Development Project of Sichuan Province (2018SZ0333), and the Innovation Team Project of Education Office of Sichuan Province (14TD0049).

**Acknowledgments:** We thank Xi Xinqiang and Hu Xiaoli for analysis of random effects.

**Conflicts of Interest:** The authors declare no conflict of interest.

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
