# Peer review of "Changes in Soil C:N:P Stoichiometry and Microbial Structure along Soil Depth in Two Forest Soils"

_forests, doi:10.3390/f10020113_

Reviewer 1 Report

The forest above the soils significantly influences the soils. The authors state in line 161 that the DBH of trees was measured. I believe that it is necessary for the to be DBH is reported, therefore Please report the data about the average DBH of the trees on the plots and at least the s.e. of those trees. Ti gives the reader a better representation of the stands, and their state.  I suggest adding the data in table 1.

Reviewer 2 Report

I would like to thank the authors for their work on the manuscript. They clearly stated their hypotheses, which gives a purpose to the paper and makes it more interesting to read. The introduction was also improved. The authors removed two markers from the PLFA analyses after previous remarks, and subsequently redid their analyses and graphs, which is commendable. However, there are still a few points that need reviewing, detailed below. I therefore recommend to the editor ‘minor revisions’.

1.       In the first review, I asked if the authors included a random effect into their statistical model, to which they responded ‘no’, without explanation. In order to successfully validate the pre-conditions of independent data to use an ANOVA, the authors would need to include a random effect to take into account the possible dependency between samples within a single core, and then within a subplot. I.e. random = ~1|Subplot/ Replicates. This point is particularly important because if the statistical analysis is incomplete, this may lead wrong results. I am not satisfied with the previous response and I strongly recommend to the authors to use the best models representing their experimental design (and the non-independence among soil samples within a given soil core).

2.       I also asked for clarity regarding the link between Figure 4 and Table 4. From my understanding of the table, it seems that the indirect effect numbers do not add up. The authors replied to the previous question saying ‘In Table 4, one indirect effect was the product of the direct path coefficients, and the total indirect effects were the summation of the whole indirect effects’. However, I tried to find the calculate the indirect effects from the figure but did not find the same values as in the table.

If we take, for example, the effect of soil type on TC :

Figure 4:

direct effect = -0.46

indirect effect = (-0.80*-0.51)+(-0.48*-0.07) = 0.408+0.0336 = 0.4416

Table 4:

direct effect =-0.462

Indirect effect = 0.390

Could the authors explain in more detail, how these indirect effects are calculated? Further, why wasn’t a co-effect considered between fungi and bacteria? I.e. why wasn’t the effect of bacteria on fungi also studying instead of only the effect of fungi on bacteria?

3.        There are missing references in the methods regarding the study sites, i.e. which dates are taken into account for the average temperatures and precipitations and where the litter and fine root biomass information comes from.

4.       There are several sentences in the discussion that need to be split up because they have several thoughts, making the discussion hard to read. For example, [L 325-329], [L387-392], [L 395-398].

5.       The conclusion needs to be revised. The stoichiometry changed with soil profile. This concludes the authors results. Also, [L 401-402] This sentence needs to be explained more clearly. Finally, the authors should have a larger implications statement. Why are the results important?

6.       Finally, I found errors in the English writing (misuse or missing plurals, phrases needing to be reordered, run-on sentences, etc) in almost every other sentence and would recommended it to be read over by an English native speaker. In several instances (i.e. the end of paragraph 2 of the introduction, at lines 60-63, and in the discussion as mentioned above), the current sentence structures do not adequately link the thoughts together.

Overall, let me reiterate that the authors satisfied most of my comments and considerably improved the clarity of the manuscript. However, I think that additional modifications are required.

A few examples from the beginning:

[L 27] indicator(s) of soil organic … to (the understanding of ) soil C

[L 27-28] and (the response) to global climate change

[L31-32] Soil C:N ratio may explain 99.2% of variation in soil dissolved organic carbon, and may therefore be used to explore…

[L 33] quantitate à quantify

[L 34] Sentence needs rephrasing

[L 38] Lots ofà many

[L 39] all items in the list need to be singular i.e. soil type, vegetation type

[L 39-40] There are a wide range of C:N:P ratios when comparing data from studies in different countries.

Author Response

Please find the response in the attachment file.

This manuscript is a resubmission of an earlier submission. The following is a list of the peer review reports and author responses from that submission.

Round  1

Reviewer 1 Report

The article is well written, and with a sound scientific foundation as well as with a high quality of presentation. The article is of great interest to the journal readers, as it focuses on the quantity of bacteria, fungi and different soils and soil properties.

Detailed comments below:

Line 44 and 45: density is expressed in kg/m2 instead of kg/m3

Line 106-107, in the table forest age (a) is written in the table, but should be only mentioned in the header

Line 106-107, I suggest adding the shrub biomass and shrub DBH into the table, as aboveground forest influences the soil and its development. 

Reviewer 2 Report

This study compares nutrients, their stoichiometry and microbial biomass in two different forest soil [Haplic luvisols and Eutric cambisols] and along a soil profile [0-10, 10-20, 20-30, 30-50, 50-100 cm]. Using the data acquired, the authors then build a structural equation model to study the direct and indirect effects of soil types and soil depth on soil water content, soil bulk density, nutrient contents, and on soil microbial biomass, separating fungi from bacteria. While the results are interesting and well interpreted, this article lacks hypothesis and a purpose to why these two soils were studied. If the authors can properly explain their choice and larger importance in studying these two soils, this paper will be of more interest to the larger forest community. 

General comments:

The structure of the article is easy to follow. In the introduction, results, and discussion, the authors always first present the soil nutrient data and then the soil microbial biomass data. The English, for the most part, is also understandable but certain sentence structures makes it hard to understand what the authors mean to say. Most of the English revisions are thus in sentence structure, with minor spelling errors. There are three major points that need to be addressed.

First, the introduction needs to be majorly revised. An introduction should not be a list of results from several articles (L38-L58), but rather a summary of major points and conclusions derived from previous studies. The introduction should ultimately give the reader sufficient background and guide the reader into the scientific question of the article. Here, the major issue with this submitted paper is that it lacks hypotheses and ultimately a reason for why the two soil types were chosen. This is not explained in the introduction and is missing in the discussion as well. In other words, why the authors study two soils in the context of their study? For instance, many studies have shown that microbial community structure and/or microbial biomass varied along land-use or environmental gradients because of plant traits and/or soil abiotic factors (e.g., Strickland & Rousk 2010 – Soil Biology Biochemistry; de Vries et al. 2012; Ecology Letters; Fanin & Bertrand 2016 – Soil Biology Biochemistry). Here, I think that this manuscript can be deeply improved by explaining to the readers why it is important to compare these two soil types in the introduction section.

Second, the methodology to determine the bacterial biomass is questionable. Here, the authors mentioned that bacteria were represented by PLFAs C13:0, C14:0, C15:0, C16:0, C18:0, 16:1ω5c, 16:1ω7c, 16:1ω9c, 18:1ω7c, 18:1ω8t, cy17:0, cy19:0, i15:0, i16:0, i17:0, i18:0, 2Me18:0, and 10:Me18:0 (L129-L131). But the markers C13:0, C14:0, C15:0, C16:0, C18:0 are not usually considered for measuring bacterial (e.g., Frostegård and Bååth 1996 – Biology and Fertility of Soils; Zelles 1999 - Biology and Fertility of Soils; Frostegård et al. 2011 - Soil Biology Biochemistry) and are often used as “general markers”. I also would like to see the reference for the marker 18:1ω8t and the marker 16:1ω5c is often used as an indicator of arbuscular mycorrhizal fungi (Olson 1999 – FEMS microbiology ecology). Furthermore, the markers a15:0 and a17:0 that are markers of gram-positive bacteria are absent although they are often abundant in a variety of soils. Finally, the authors cite 4 studies for justifying their choices, but none of them have used the general markers C13:0, C14:0, C15:0, C16:0, C18:0 cited above as indicator of microbial biomass. For all these reasons, I think that the authors should recalculate properly the bacterial biomass, because this can have important consequences on their results, and thus on their conclusions.

Third, it is not clear between Figure 4 and Tables 3 and 4 from where the standardized path coefficients are calculated and inserted into the structural equation modeling. It is important to clear this up in order to ensure that the conclusions derived from this model are clear and straightforward. Moreover, how the indirect effects have been calculated? It is not clear in the Table 4 and more information are necessary for the readers. In particular, the results section 3.4 should be heavily revised.

Detailed comments:

[L18]’while the effects of soil types were not significant’. Confusing considering in line [14] you say that C:N ratio is significantly higher in Eutric cambisols than Haplic luvisols

[L20] “TC” à total carbon

[L22] The abstract needs an overall conclusion. The current last sentence only concludes the previous results sentence

[L26] Phosphorus

[L92] “chosen in Dulan County’ à why were they chosen?

[L98] The list describing soils at B layer is confusing. Does this apply to all B layer soils?

[L112] “five soil samples were collected” à were these five samples at each layer in one core, or were five cores samples, with one sample per layer?

[L117] “five cutting rings”à at the same five layers?

[L134] What kind of model was used? Was there a random effect taking into account correlations between depths of a single core?

[L158] “The same following” à what does this mean?

[L156] specify the relation between * and **, with respective p-values

[L179] Figure 3 : if the purpose is to relate the biomasses between the soils, consider added them both on the same graph. The current difference in scales makes both biomasses in both soils look very similar in quantity.

[L189] “variable SMB explained 92% and 60% of the variation in the variable…” à where do these data come from? Unclear if from Figure 4 or one of the tables.

[L193] “Percentages” à R^2 values are not percentages (i.e. 0.67 is not a percent; 67% is a percent)

[L200] very unclear where is table 4 the following percentages come from: 40 %, 50% and 43%

[L217] Sentence should start with “this may explain why the soil water content…” instead of therefore. The structure of the first paragraph of the discussion seems like the results were obvious. Should rather focus on explaining results based on background (which is quite interesting).

[L225-L227] Sentence “The SBD of Haplic Luvisols…” should be discussed. Lacking a reason for this increase and then decrease of coil C:N. Why is there a link between the two?

[L231-L232] “resulted in a more rapid decomposition rate of soil organic matter and more P contents were released” à could this also apply to N?

[L242-L243] “release more soil P because of quicker soil mineralization’ à why not also release more soil N?

Reviewer 3 Report

This study investigated C, N and P concentrations, their ratios and associated microbial communities at different depths in two forest soils. The work is competent and adds to the general knowledge in this area but the outcomes are more or less what I would expect.

There are multiple (mostly but not all) minor issues of language which mostly allow the reader to comprehend what is meant but in some cases lead to a loss of clarity. The text could really do with editing by a native english speaker (other than me!)

On a few specific point;

Not sure I understand the last line in the Abstract.

I cannot reconcile the description of horizon depths with those of the sampling programme.

The indicated size of plots is inconsistent betyween Table 1 and text.

Lines 117 - 119 and not comprehensible.

Intervals in the y axes of figure 1 could be more detailed.

L 158 'The same following' ??

Some of the links in the SEM are pretty inevitable e.g. SMB vs funi + bacteria.

I found the text in lines 202 - 208 difficult to follow.

The same in sections of 232 - 243